



# Do large-scale wind farms affect air quality forecast? Modeling evidence in Northern China

Si Li[1,3], Tao Huang[1]*, Jingyue Mo[4], Jixiang Li[1], Xiaodong Zhang[1], Jiao Du[1], Shu Tao[2],
Junfeng Liu[2], Wanyanhan Jiang[1], Lulu Lian[1], Hong Gao[1], Xiaoxuan Mao[1], Yuan Zhao[2],
Jianmin Ma[2, 1]*

[1]Key Laboratory for Environmental Pollution Prediction and Control, Gansu Province; College of Earth
and Environmental Sciences, Lanzhou University, Lanzhou 730000, P. R. China

[2]Laboratory for Earth Surface Processes, College of Urban and Environmental Sciences, Peking
University, Beijing 100871, P. R. China

[3]College of Atmospheric Sciences, Lanzhou University, Lanzhou 730000, P. R. China

[4]Chinese Academy of Meteorological Sciences, Beijing 100000, P. R. China

*Corresponding to: Jianmin Ma (jmma@pku.edu.cn) and Tao Huang (huangt@lzu.edu.cn)

**Abstract.** Wind farms have been found to alter local and regional meteorology and climate. Here, we
show that multiple large-scale wind farms might disturb air quality forecasts and affect $PM_{2.5}$ air pollution.
We explore the impact of large-scale wind farms on $PM_{2.5}$ concentrations and forecasts in the Northern
China Plain in winter and summer using a coupled weather forecast - atmospheric chemistry model
(WRF-Chem). Modelling results reveal that the large-scale wind farms decrease $PM_{2.5}$ levels within the
wind farms and increase $PM_{2.5}$ concentrations by 49% and 16% of the modelled monthly mean $PM_{2.5}$
concentrations in proximate areas and regions hundreds of kilometres downstream. The wind farm-forced
changes in $PM_{2.5}$ are more evident in the simulated hourly $PM_{2.5}$ concentrations. The model sensitivity
studies reveal that hourly concentration fractions in winter induced by wind farms vary from -40% to
250% in nearby and distant downstream regions and metropolises, comparing with the cases without the
wind farms. The impact of wind farms on modeled $PM_{2.5}$ during the nighttime is stronger than that in the
daytime. Our results suggested that the wind farm perturbed changes in $PM_{2.5}$ should not be overlooked
because such changes might affect air quality forecast on an hourly basis, particularly in heavily
contaminated Beijing-Tianjin-Hebei region by $PM_{2.5}$.

## 1 Introduction

To reduce the emissions of greenhouse gases and air pollutants that resulted from fossil fuel
combustion during the course of industrialization and urbanization, China has launched and set a low-



carbon energy development strategy and goals since the 2000s. Wind power is one of the leading clean

energy technologies and has rapidly developed (Mo et al., 2017; McElroy et al., 2009). According to the

Thirteenth Five-Year Plan issued by the Chinese National Energy Administration (CNEA, 2016), wind

power has become the third largest power source in China after coal and hydroelectric power. In 2017,

the total installed capacity of wind power in China reached 188 GW. Presently, there are nine 10-million

kW class giant wind power bases in northern China, extending from Xinjiang in the west to northeast

China. By the end of 2020, the total installed capacity of wind power will reach 210 GW, which will

account for 6% of the country's total power generation (CNEA, 2016).

Large-scale wind farms can alter the turbulence field near an underlying surface by changing the

surface roughness lengths and spinning wind turbine rotors. Extensive field measurement and modelling

studies have revealed that wind turbines could significantly reduce wind speed at the wind turbine hub

height within and downwind of wind farms and generate turbulence in turbine wakes (Baidya Roy, 2004;

Christiansen and Hasager, 2005; Fitch et al., 2012). The increasing turbulent strength and activities

induced by wind turbines cause nocturnal land surface temperature warming, enhance the vertical mixing

of momentum, heat, moisture and other scalars such as air pollutants, and increase the boundary-layer

height (Porté-Agel et al., 2014; Li et al., 2018; Fitch et al., 2013; Baidya Roy and Traiteur, 2010; Frandsen

et al., 2006). Satellite synthetic aperture radar (SAR) has discerned a reduction in the mean wind speed

of 8-9% in a giant wind farm (Christiansen and Hasager, 2005; Jimenez et al., 2015). The large-scale

wind farms also exert dynamic and thermodynamic impacts on downstream areas via a "downstream

impact" (Sun et al., 2018; Barrie and Kirk-Davidoff, 2010) and wake turbulence (Fitch et al., 2012).

The present study assesses the wind farm-induced changes in air pollutant redistribution and

forecasts over all of northern China, where many giant- and large-scale wind farms have been built in

the past two decades. **Figure 1** illustrates these wind farms in northern China. As seen, intensive wind

farms are located in Inner Mongolia and northern Hebei Province to the north of the Northern China

Plain (NCP), including Beijing-Tianjin-Hebei, one of the regions in China and the globe most heavily

polluted by $PM_{2.5}$ and surface ozone ($O_3$) due to rapid industrialization and urbanization in the last three

decades and the large capacity of the steel and energy industries in this region (Lu et al., 2018; Tao et al.,

2017). Although wind energy plays an increasingly important role in the sustainable development and

reduction of air pollutant emissions, it is not clear if and to what extent these large-scale wind farms

(hereafter referred to as the wind farm chain (WFC) in northern China) could disturb the cycling, fate,



and forecasts of air pollutants in the NCP. It is important to fill this knowledge gap because the perturbations of atmospheric transport of air contaminants induced by WFC could potentially affect regional air quality forecasts and the effectiveness of air pollution mitigation measures.

## 2 Materials and Methods

### 2.1 Model configuration

**Figure 1** illustrates the nested model domain and the major large-scale wind farms (WFC) in northern China, highlighted by red colour shading. In particular, the WFC is located upstream of the NCP subject to the East Asia winter monsoon featured by prevailing westerly and northerly winds (**Figure S1a**) or downstream subject to the East Asia summer monsoon with prevailing southeasterly winds (**Figure S1b**). The nested model domain in the Lambert projection (Bossioli et al., 2016) was set with

the outer domain (d01) covering northern China, extending from 88° to 132° E and from 29° to 50° N at a resolution of 50 km × 50 km. The inner domain (d02) with a resolution of 10 km × 10 km, covers a large area of Inner Mongolia, the NCP, and its surroundings. The total number of wind turbines in the outer domain (northern China, **Figure 1**) was approximately 81,000. Considering the nominal power of each turbine as 1.5 MW, these wind turbines yield the total power capacity of 121,500 MW in northern

China, accounting for about 70% of total wind power capacity in China in 2016 (Global Wind Energy Council (GWEC), 2017). According to the GWEC statistics, the power capacity in northern China's windfarms was 72% of its total in entire China but this percentage decreased in 2016 due to increasing wind abandoning rate in northern China in 2016. It is noted that the 10 km × 10 km horizontal resolution is greater than the distance between wind turbines in the WFC. In previous modelling studies for the

effect of wind farms on local meteorology, some investigators set 1-3 km horizontal spacing for a single wind farm (Fitch et al., 2012; Baidya Roy et al., 2004). For multiple large-scale wind farms, however, the model horizontal resolutions were chosen as several ten kilometres (Baidya Roy et al., 2010; Vautard et al., 2013; Sun et al., 2018). In the evaluation of wind farm parameterization in WRF model, Lee and Lundquist (2017) found that WRF performance did not depend on the number of wind turbines per model

grid cell and array of wind turbines. In our case, the 10 km spatial resolution could be appropriate with the wind farm parameterization schemes as will be elaborated below.

The coupled weather forecast and atmospheric chemistry model WRF-Chem (version 3.8.1) was



applied in our numerical investigations. There are 29 vertical model levels, with the top of the model

atmosphere at 50 hPa. To better simulate the effects of a wind farm on the ABL and air pollutants, higher

vertical resolutions within the atmospheric boundary layer (ABL) and the surface boundary layer (SBL)

(approximately 5, 10, 50, 80, 100, 120, 180, 300 m, and so on) were adopted in the present study. The

main physical parameterization options implemented here are presented in **Table S1**. The chemical

mechanism used MOZART Chemistry with the MOSAIC 4-bin aerosol scheme using the KPP library

(Ackermann et al., 1998; Borbon et al., 2013).

Meteorological data for the initial and boundary conditions of WRF-Chem were obtained from the

National Center for Environmental Prediction (NCEP) reanalysis on a spatial resolution of 1°×1°

latitude/longitude   (https://www.esrl.noaa.gov/psd/data/gridded/reanalysis).   The   anthropogenic

emissions used the MIX regional emission inventory (Li et al., 2017). Measured hourly winds and air

temperature data at the observational stations in the inner model domain (**Figure 1**) for the evaluation of

WRF modelled meteorology were collected from NOAA's National Climatic Data Center (NCDC),

available at ftp://ftp.ncdc.noaa.gov/pub/data/noaa/isd-lite/.

The MIX emission inventory provides the gridded air emission data of $SO_2$, $NO_x$, CO, NMVOC

(non-methane volatile organic compounds), $NH_3$, $PM_{10}$, $PM_{2.5}$, BC, OC, and $CO_2$ on a 0.25° × 0.25°

latitude/longitude resolution. The biogenic emissions were collected from Model of Emission of Gases

and Aerosols from Nature (MEGAN) (Guenther et al., 2012) at a spatial resolution of approximately 1

km. The emissions from biomass-burning were obtained from the Fire Inventory from NCAR (FINN)

program at a resolution of 1 km (Wiedinmyer et al., 2011).

A contrast experiment, $t$ test formulation was used to test the significance of the modelled $PM_{2.5}$

differences and fractions subject to the different model scenarios. Following Sun *et al*. (2018), the

contrast experiment $t$ test is written as

$$t = \frac{\bar{d}}{SD/\sqrt{n}} ,$$   (1)

where $\bar{d} = \frac{1}{n}\sum_{i=1}^{n}\Delta PM_{2.5i}$, $\Delta PM_{2.5}$ is the hourly difference between two model scenarios, $n$ is the

number of hours, and SD is the standard deviation. Note that the T-test was also applied in $PM_{2.5}$ fractions

between the WFC-related model scenarios and the BASE scenario but replaced the difference $\Delta PM_{2.5}$ by

$PM_{2.5}$ fractions, calculated by, for example, $(C_{Si} - C_{BASE})\times100/C_{BASE}$, where $C_{Si}$ is the $PM_{2.5}$

concentrations from any of the WFC-related model scenario simulations and $C_{BASE}$ is the $PM_{2.5}$



concentration from the BASE run.

**2.2 Wind farm parameterization**

Detailed information about the wind farms in northern China was collected from the China Certified

Emission Reduction Exchange Info-Platform (CCER, http://cdm.ccchina.org.cn/ccer.aspx) and Google

maps, from which the locations of wind farms and wind turbines in northern China were collected

(**Figure S2**). Based on these data, we set the average hub-height of wind turbines at 96 m, the blade (rotor)

diameter as 113 m, and the average wind turbine spacing at 678 m (approximately six times the rotor

diameters). We used two wind farm parameterization schemes in our modelling. The first is the increased

roughness length parameterization, which specifies the wind turbines by an increase of the surface

roughness (Mo et al., 2017; Keith et al., 2004; Ivanova and Nadyozhina, 2000). This parameterization

scheme calculates the aerodynamic roughness length $Z_0$ (m) of a wind farm, defined by as (Lettau, 1969):

$$Z_0 = 0.5h^* \frac{s}{S} ,\qquad\qquad(2)$$

where $h^*$ is the average vertical extent (m) or hub height for wind turbines and $s$ is the silhouette area

(km$^2$) of the average obstacle or the rotating area for wind turbine blades. $S$ is the density of roughness

elements, calculated by $S=A/n$, where $A$ is the total area occupied by obstacles (km$^2$) and $n$ the total

number of obstacles. As aforementioned, the wind turbine spacing is set as six times the diameter of the

wind turbine rotor. By setting s and $S$ as 0.01 km$^2$ and 0.46 km$^2$, we obtained the roughness length $Z_0$ of

1.04 m. This method has the advantages of simplicity and accuracy in representing the wind turbines

(Wieringa, 1993; Petersen, 1997).

The second scheme, developed by Fitch et al. (2012), considers the wind turbines as a momentum

sink within the turbine rotor area, which transforms kinetic energy (KE) into electricity and turbulent

kinetic energy (TKE, m$^2$ s$^{-2}$). This scheme extracts the total fraction of KE from the air utilizing a thrust

coefficient $C_T$, which is turbine type dependent and the function of wind speed. A proportion KE is

converted to electrical energy by a power coefficient $C_P$. The rest of KE is converted to TKE, defined by

the TKE coefficient $C_{TKE}=C_T-C_P$. The $C_T$ and $C_P$ can be taken from the turbine manufacturers. The WRF

model implements the Fitch scheme (Fich et al., 2012, 2015; Yuan et al., 2017) which accounts for the

effects of local wind drag on wind-energy extraction and power estimation (Lee and Lundquist, 2017).

This scheme has been extensively applied in windfarm-meteorology interactive modelling using WRF

model (Cervarich et al., 2013; Xia et al., 2016, 2017; Fiedler and Bukovsky, 2011; Vautard et al., 2013;





Mo et al., 2017; Yuan et al., 2017; and Sun et al., 2018), and was also adopted here with a standing minimal thrust coefficient of 0.16. However, Volker et al. (2012) have reported that Fitch-scheme estimated thrust applied to the flow was overestimated by almost one order of magnitude and that the modeled TKE in the Fitch-scheme diffused the velocity deficit deep into the boundary layer, and caused unnaturally high positive velocity deficits at the lower boundary.

If the turbines are assumed to be oriented perpendicular to the wind flow, the drag force created by wind turbines is defined by:

$$F_{drag} = \frac{1}{2} C_T \rho V^2 A \ , \tag{3}$$

where $\rho$ is the air density, $A=(\pi/4)D^2$ is the cross sectional rotor area (D is the diameter of the turbine blades), and V(u, v) is the horizontal velocity.

The rate of loss for KE from the air subject to a wind turbine reads:

$$\frac{\partial KE_{drag}}{\partial t} = -\frac{1}{2} C_T \rho V^3 A \ , \tag{4}$$

The electric energy extracted from KE is defined by:

$$\frac{\partial P}{\partial t} = \frac{1}{2} C_P \rho V^3 A \ , \tag{5}$$

and the TKE extracted from the rest of KE is given by:

$$\frac{\partial TKE}{\partial t} = \frac{1}{2} C_{TKE} \rho V^3 A \ , \tag{6}$$

where the nominal power of a turbine is taken as 1.5 MW.

To examine the sensitivities of air quality in the NCP to WFC, we conducted extensive model simulations of PM$_{2.5}$ in the NCP subject to four model scenarios (**Table S2**). The first model scenario is the control run (scenario 1, S1), in which the WFC was not taken into consideration, hereafter referred to as the BASE or no-WFC simulations. The second and third model scenarios used the surface roughness length parameterization scheme (referred to as the SRL run, S2) and the drag force parameterization (referred to as the DFP run, S3). Considering the future wind farm development in northern China, in the fourth model scenario we projected a double area of WFC with the drag force parameterization from the current wind farm area as shown in **Figure 1** (referred to as the DOU run, S4). WRF-Chem was run for January and July 2016 to examine the responses of PM$_{2.5}$ to different setups of the model scenarios under typical winter and summer atmospheric circulations.

**2.3 Model evaluation**



Monitored PM$_{2.5}$ concentration data at the selected locations in January and July 2016 were used to

verify the WRF-Chem modelling results. We selected hourly in situ PM$_{2.5}$ measurement data at five

sampling sites in the model evaluation, collected from the national air quality automatic monitoring

stations in China (http://106.37.208.233:20035/), including Beijing, Chengde, Hingdan, Siping, and

Guyuan. Detailed model evaluations against the monitoring data at these sampling sites are presented in

Supporting Information (**SI**) **Text 1.1, Figures S3-S5, Tables S3-S5**. The WRF predicted winds and

temperature were also evaluated against measured data. Results are presented in SI **Text 1.2**, **Figures S6-**

**S9**, and **Tables S6-S11**.

### 3 Results

### 3.1 WFC disturbed hourly and daily PM$_{2.5}$ in winter

**Figure S10a** shows WRF-Chem-simulated monthly averaged daily air concentrations of PM$_{2.5}$ from

the BASE run (S1, Methods) at the lowest model level (~5 m above the ground surface) across the model

domain in January 2016. High concentrations can be observed in the NCP due to strong emissions of

PM$_{2.5}$ precursors (*e.g.*, sulphur dioxide, SO$_2$, and nitrogen oxide, NO$_x$) in these heavy industrial regions

featured by steel, energy, and cement industries (Zhao et al., 2013; Xu et al., 2016; Wang et al., 2018).

**Figure S10b-d** illustrates the concentration differences between model scenarios 2-4 simulations and the

BASE run (S1) results. Marked positive PM$_{2.5}$ differences occur in central and northeastern Hebei

Province from all three model scenario runs. The maximum PM$_{2.5}$ difference was as high as 14 μg m$^{-3}$ in

the highly polluted Beijing-Tianjin-Hebei region, which was a maximum 10% increase in monthly mean

PM$_{2.5}$ concentration. This is also illustrated by the fraction of mean PM$_{2.5}$ concentrations from the SRL

(S2) and DFP (S3) runs to that from the BASE run (S1) in **Figure 2a** and **2b**, estimated by $f_i = (C_{Si} - C_{BASE})$

$\times 100/C_{BASE}$, where $C_{Si}$ is monthly mean PM$_{2.5}$ concentration from the SRL (i=1) and DFP (i=2) runs and

$C_{BASE}$ is PM$_{2.5}$ concentration from the BASE run, respectively. The results from the two wind farm

parameterization schemes do not exhibit significant differences in simulated PM$_{2.5}$ concentrations. The

negative concentration fractions can be identified in Inner Mongolia in the northern WFC. For the DOU

run, doubling the wind farm installation expands the area with positive concentration differences (**Figure**

**S10d**), which extends from Beijing-Tianjin-Hebei to Shandong Province, although the values of the

positive fractions were larger than those from the SRL and DFP runs (**Figure 2a, 2b**). It is also evident



that the doubling of the wind farm area (DOU run) resulted in the expansion of areas with negative $PM_{2.5}$ fractions (**Figure 2c**) compared with the SRL and DFP simulations (**Figure 2a** and **2b**), which predicts lower $PM_{2.5}$ concentrations in the vicinity of the WFC. However, as shown in **Figure S11**, which

illustrates the fractions of modelled $PM_{2.5}$ daily air concentrations in January 2016 from the three model scenarios in Zhangjiakou near the WFC, doubling WFC area enhances $PM_{2.5}$ concentrations by 30%-150%, which generally indicates a positive response of $PM_{2.5}$ to the expansion of wind farm areas.

**Figure S12** illustrates WRF model-simulated mean wind speed differences between the WFC-related scenarios and the BASE scenario at the hub height averaged over January 2016. It has been known

that wind turbines act to extract momentum from the atmosphere and produce large wind speed deficits within and around wind farms at the hub-height level. Such deficits can be clearly identified from wind speed differences between the WFC-related simulations (SRL, DFP, DOU) and the BASE run. Negative differences of the mean wind speeds with and without considering wind farms occur within and around wind farms (**Figure S12a-c**), which indicates declining wind speeds. The weak wind speeds are

significantly associated with the locations of the WFC, as shown in **Figure S12**. The maximum wind speed reduction reached 6 m s$^{-1}$ within the wind farm, which suggests a 40% reduction of wind speed. The deficit of wind speeds gradually diminishes in the downstream (south) of the WFC. Since the lower wind speeds result in weak atmospheric transport and diffusion of air pollutants, the simulated $PM_{2.5}$ tends to increase in the downstream of the WFC, as shown in **Figure S10b-d** and **Figure 2a-c**. Mo *et al*.

(2017) proposed an edge effect of a large-scale wind farm on the spatial distribution of air pollutants within and around the wind farm, featured by a higher concentration at the immediate upwind and border region of the wind farm. This edge effect was attributed to the changes in wind speed and turbulence intensity driven by the rotation of the rotor blades and the increase of the effective surface roughness length within the wind farm, which led to a step change from the smooth (upstream region) to the rough

(wind farm) underlying surface (Mo et al., 2017; Barrie and Kirk-Davidoff, 2010). In the downstream region of a wind farm, the air pollutant concentration tends to decrease, induced by increasing wind speed due to the rough-to-smooth underlying surface transition and recovered momentum in the downstream of the wind farm from the lost momentum within the wind farm. However, due to relatively lower friction velocity and weak turbulence over the smooth surface, the decreasing turbulent mixing could reduce

$PM_{2.5}$ concentration vertical mixing and thus enhance the $PM_{2.5}$ level near the surface. This mechanism, together with the downstream wind speed deficit, would result in an overall increase of $PM_{2.5}$


concentrations in downstream (**Figure S10** and **Figure 2**).

**Figure 3a** displays the fractions of WRF-Chem-modelled hourly $PM_{2.5}$ concentrations ($\mu g\ m^{-3}$) in January 2016 from the SRL simulation to that from the BASE run in northeast (NHB) and central Hebei

(CHB) Province, Zhangjiakou City (ZJK), Beijing (BEJ), and Tianjin (TIJ) (**Figure 1**). Among these regions and cities, northeast Hebei and Zhangjiakou are proximate to the WFC, central Hebei is a mostly contaminated region by $PM_{2.5}$, and Beijing and Tianjin are two megacities that were also subject to severe $PM_{2.5}$ contamination in 2016. Strong perturbations of $PM_{2.5}$ induced by the WFC on an hourly basis can be discerned in NHB, with the maximum fraction of up to 250% (green dashed line), followed by ZJK

with the maximum fraction of approximately 180% (blue dashed line), which confirms that the stronger influences of the WFC on $PM_{2.5}$ distribution occurred in its proximate locations. Large fractions of modelled $PM_{2.5}$ hourly concentrations can also be identified in central Hebei (CHB), with the maximum $PM_{2.5}$ fraction exceeding 170% on January 12, 2016 (purple dashed line, **Figure 3a**). In the two megacities of Beijing (red solid line) and Tianjin (black dotted line), the WFC-disturbed maximum $PM_{2.5}$

hourly concentration fractions also ranged from -24% to 165%, and in many occasions, the concentration fractions fluctuated by ±30%, which manifested marked concentration perturbations in these two megacities induced by the WFC, although the two megacities are 200-300 kilometres away from the WFC. The monthly average hourly concentration fractions were 49% for northeast Hebei, 16% for central Hebei, 5% for Beijing, 0% for Tianjin, and 3% for Zhangjiakou, which again shows significant responses

of modelled $PM_{2.5}$ perturbations to the WFC in downstream regions and cities. Different from positive and negative hourly concentration fluctuations, the monthly concentration fractions were positive in all selected regions and cities. The influences of the WFC on $PM_{2.5}$ in Beijing (5%) and Tianjin (0%) were less significant on a monthly basis, potentially within a model error range. Although the $PM_{2.5}$ hourly concentration fractions at Zhangjiakou showed large fluctuations next to the NHB, the monthly mean

concentration fraction was small (**Figure 3b**). The large fractions of simulated $PM_{2.5}$ hourly concentrations with and without the presence of the WFC seem to suggest that the WFC exerts a strong influence on hourly changes in downstream $PM_{2.5}$ concentrations, which causes uncertainties in $PM_{2.5}$ forecasts in these regions and cities. Such significant influence even extends to the monthly average $PM_{2.5}$ concentrations by as much as 50%.

Hourly or diurnal changes in local atmospheric circulation and wind and turbulence fields could play an important role in $PM_{2.5}$ concentration fluctuations. We selected two cases in Zhangjiakou, with





the maximum concentration fraction (192.7%) at 0700 local standard time (LST) January 4, 2016 and

the minimum concentration fraction (-37.3%) at 1500 LST January 16, 2016, to examine the associations

between fluctuations of the $PM_{2.5}$ concentration fractions and local atmospheric circulation. Results are

illustrated in **Figure 4**, which illustrates the model-simulated fractions of $PM_{2.5}$ concentration between

the DFP and BASE model scenario simulations near the surface. Positive concentration fractions

downstream of the WFC when the maximum concentration fraction occurred at 0700 LST on January 4,

2016 (**Figure 3a**) and minimum concentration fraction within and downstream of the WFC at 1500 LST

in January 16, 2016, the time that the minimum $PM_{2.5}$ fraction occurred (**Figure 3a**). We further examined

the wind speed differences at the hub height in these two cases. Located in the winter monsoon regime,

the northwesterly wind prevails over the model domain (**Figure S1**). In these two cases subject to the

maximum and minimum $PM_{2.5}$ concentration fractions, the modelling results show larger negative wind

speed fractions (**Figure 4d**) within and around the WFC in the case of the minimum $PM_{2.5}$ fractions that

occurred at 1500 LST in January 16, 2016, which indicates a stronger wind speed deficit or decline in

the presence of WFC compared with the no-WFC case (BASE scenario), whereas the smaller negative

wind speed fractions occurred in the case with the maximum $PM_{2.5}$ fraction (**Figure 4c**). The larger

negative wind speed fraction in the minimum $PM_{2.5}$ concentration fraction than that in the maximum

$PM_{2.5}$ concentration fraction was not expected because the decreasing wind speed should have resulted

in higher $PM_{2.5}$ concentration. We would thus expect positive $PM_{2.5}$ concentration fraction in **Figure 4b**.

**Figure 4c** and **4d** also illustrate wind vectors superimposed with wind speed fractions, from which we

could identify a southerly flow extending from the NCP to the WFC in the case of the maximum $PM_{2.5}$

concentration fraction (**Figure 4c**), although the northwesterly wind prevails in the winter season in this

region. This southerly flow could provide an atmospheric transport route delivering $PM_{2.5}$ from the

mostly contaminated NCP region to the WFC. However, in the case of the minimum $PM_{2.5}$ fraction we

can observe relatively strong northwesterly and northeasterly winds across the WFC, which do not favour

the atmospheric transport of $PM_{2.5}$ from the south of the WFC as the major source region of $PM_{2.5}$

precursors. In this sense, the regional wind field plays a more important role than the WFC-perturbed

wind field.

Figure 4e and 4f show the fractions of the simulated turbulent kinetic energy (TKE) ($m^2 s^{-2}$) at the

hub height from the DFP model scenario to that from the control (BASE) run. The WFC-related model

scenario generated stronger TKE than the no-WFC case, characterized by positive TKE fractions.



Comparing TKE fractions between the two cases, one can identify stronger fractions within and around the WFC on January 16, 2016 (**Figure 4d**), when the minimum $PM_{2.5}$ concentration fraction occurred. Since wind turbine extracts energy from the atmosphere and creates turbulence in its wake, larger TKE

occurs in the downwind region of the WFC, which enhances the vertical mixing of $PM_{2.5}$ and thereby reduces $PM_{2.5}$ levels near the surface. As a result, the larger negative $PM_{2.5}$ concentration fraction corresponds to a greater positive TKE fraction, as shown in **Figure 4b** and **4f**. The downstream undershooting from the step changes in the surface roughness might also contribute to the declining $PM_{2.5}$ or larger negative $PM_{2.5}$ concentration fractions. Under the northerly wind regime, the rough-to-smooth

underlying surface transition from the WFC to its downstream could increase wind speed and accelerate the southward transport of $PM_{2.5}$, thus reducing $PM_{2.5}$ concentration in the downstream region, as shown in **Figure 4b** and **4f**.

### 3.2 WFC disturbed hourly and daily $PM_{2.5}$ in summer

**Figure 5** shows WRF-Chem-simulated monthly mean $PM_{2.5}$ air concentrations in July 2016 in

northern China and the mean concentration differences between the control model (BASE) run and SRL, DFP, and DOU model scenario runs in July 2016. Similar to the winter case, severe $PM_{2.5}$ pollution occurred in the Beijing-Tianjin-Hebei area. Positive $\Delta PM_{2.5}$ can be discerned in the south (downstream) of the WFC, which is more evident in the DFP scenario run (**Figure 5c**). Doubling the WFC seems not to significantly alter the monthly $\Delta PM_{2.5}$ (**Figure 5d**). Note that positive $\Delta PM_{2.5}$ can also be observed in

the far south of the WFC such as in Shandong Province and over the coastal waters of the Bohai Sea and the Yellow Sea. It is not yet clear how the WFC could force increasing $PM_{2.5}$ air concentrations in those places far distant from the WFC. A previous study has revealed that the perturbed weather response to a large wind farm array could extend to several thousand kilometres downstream (Barrie and Kirk-Davidoff, 2010). Overall, the monthly $PM_{2.5}$ concentration fractions from the SRL, DFP, and DOU model

runs to that of the control run are small. Compared with the January case (**Figure 2**),no statistically significant positive concentration fractions were identified in the downstream of the WFC in July. This is likely attributed to stronger local boundary layer circulation occurring in the summertime. Although the summer Asian monsoon is a primary summer atmospheric circulation pattern in the inner model domain, local weather conditions are often dominated by local atmospheric circulations driven by non-

uniform surface heating and cooling (Garratt, 1994). Comparisons of the modelled monthly mean wind



speed differences between three WFC runs and the BASE run in **Figure S13** show smaller areas of negative mean wind differences in July than January, indicating that the wind deficits induced by the WFC in July is less significant than January. Differing from ΔTKE as shown in **Figure 4e** and **4f** in January 2016, modelled ΔTKE fractions (%) at the hub height between the DFP and BASE model

scenario simulations at the time of the maximum $PM_{2.5}$ fraction (0700 LST July 16, 2016) and minimum fraction (0900 LST July 14, 2016) do not illustrate marked increase within and around the WFC. Summer local circulation might disturb the WFC-induced changes in turbulence and wind speed.

Hourly and monthly concentration fractions of the SRL run to that of the control (BASE) run in the selected five regions and cities are illustrated in **Figure 6**. Compared with the winter case (**Figure 3a**),

the hourly concentration fractions do not significantly fluctuate (**Figure 6a**), which suggests that during most of July, the WFC did not frequently disturb hourly changes in $PM_{2.5}$ concentrations. However, we can still identify strong oscillations, with the maximum amplitude as high as 400% in the hourly $PM_{2.5}$ concentrations fractions at 2100 LST July 15, 2016 in NHB and 1400 LST July 24, 2016 in ZJK, which is stronger than that in the wintertime. The monthly averaged concentration fractions are all positive in

the five selected regions and cities (**Figure 6b**), with the highest fraction of 11.9% in ZJK, followed by 10.5% in NHB, 5.1% in Tianjin, 5% in Beijing, and 4.3% in CHB, which again indicates that the WFC increases $PM_{2.5}$ concentrations in these places. As shown in **Figure 6b**, the prevailing wind in northern China in summer consists of mainly southerly and southeasterly winds under the East Asian summer monsoon regime. The wind direction alters more frequently in summer than in winter. The average wind

deficit within the WFC is approximately 2 m s$^{-1}$ and 1 m s$^{-1}$ around the WFC. These wind speed deficits are much smaller than that in winter but are still visible, which likely results in weak fluctuations of the modelled $PM_{2.5}$ concentrations.

### 3.3 Daytime and nighttime PM2.5 perturbed by WFC

Baidya Roy and Traiteur found that a wind farm exerted a significant impact on atmospheric stability

and the development of the boundary layer over the wind farm, with obvious diurnal variations (Baidya Roy and Traiteur, 2010). **Figure 7a** and **7b** display monthly average daytime and nighttime $PM_{2.5}$ air concentrations in January 2016 across the model domain at 0700-1800 LST (daytime) and 1900-0600 LST (nighttime), respectively. Similar to **Figure 2b**, higher $PM_{2.5}$ levels in both day and night were simulated in the Beijing-Tianjin-Hebei area as the major source region of $PM_{2.5}$ precursors in China. A



close look at the figure one can identify higher monthly mean concentration during the nighttime than in the daytime due to weak winds and stable boundary-layer conditions. **Figure 7c-h** illustrate the fractions of monthly mean $PM_{2.5}$ concentrations ($\Delta PM_{2.5}$) between the WFC-related scenario runs (SRL, DFP, DOU) and the no-WFC scenario run (control run) during the daytime (**Figure 7c-e**) and the nighttime (**Figure 7f-h**), respectively. As shown, the WFC more strongly increases the mean $PM_{2.5}$ concentrations,

by up to 4 µg m$^{-3}$ in the nighttime in its downstream region, featured by larger and positive $\Delta PM_{2.5}$. Note that the most significant changes in $\Delta PM_{2.5}$ or increasing $PM_{2.5}$ occur in the most severely $PM_{2.5}$-contaminated region in Beijing-Tianjin-Hebei in both daytime and nighttime, rather than in the immediate neighbouring region of the WFC.

**3.4 $PM_{2.5}$ in projected WFC expansion**

The response of $PM_{2.5}$ to the projected WFC expansion can be further elucidated by comparing the simulated $PM_{2.5}$ concentration fractions between the DFP and DOU scenario runs and the BASE run in Zhangjiakou because the DOU run was performed using the same wind farm parameterization scheme as that of the DFP run (drag force parameterization) and Zhangjiakou is the city nearest to the WFC. **Figure 8** displays the hourly time series of $PM_{2.5}$ fractions of DOU (red dash line) and DFP (solid deep

blue line) to the BASE simulations in Zhangjiakou in January 2016, estimated by ($PM_{2.5DOU}-PM_{2.5BASE}$)/$PM_{2.5BASE}\times100$ (also applicable for the DFP case by replacing $PM_{2.5DOU}$ by $PM_{2.5DFP}$), where $PM_{2.5DOU}$, $PM_{2.5DFP}$, and $PM_{2.5BASE}$ are the $PM_{2.5}$ concentrations from the three model scenario simulations. In most cases, we observed large positive concentration fractions between the DOU and BASE runs, which indicates that doubling the WFC installation increases $PM_{2.5}$ levels in Zhangjiakou. In many cases

the concentration fractions between the DOU and BASE runs are considerably greater than those between the DFP and BASE runs. Note also that the fractions between the DOU and BASE simulations are often positive, whereas the DFP to BASE fractions are negative, which suggests that doubling the WFC turns the decreasing $PM_{2.5}$ concentrations to an increasing trend. The monthly average hourly fraction between the DFP and BASE scenario runs is -12.0%, and the mean hourly concentration fraction between DOU

and BASE is 12.4%. Located in the south of the WFC, Zhangjiakou is a downstream site in terms of the prevailing northwesterly winds during the wintertime. The increasing wind speed and momentum due to the downstream edge effect (rough to smooth underlying surface) could enhance the dispersion of $PM_{2.5}$ and reduce concentration in Zhangjiakou City, characterized by the negative fraction (-12% for monthly



mean fraction). Doubling the WFC area virtually encircles Zhangjiakou City within the expanded WFC,

where increasing terrain surface roughness and wind turbine results in a wind speed reduction (wind deficit) that favours increasing $PM_{2.5}$ concentration, particularly in the wake area of the WFC, characterized by the positive $PM_{2.5}$ fractions.

However, in the summertime the monthly concentration fractions between the DOU and BASE model scenario simulations are lower than those of the DFP to BASE scenario simulations, which

suggests that the expansion of WFC reduces $PM_{2.5}$ concentrations in Zhangjiakou. This is because the doubled WFC area encircled Zhangjiakou within the WFC. Under the prevailing southeasterly summer monsoon (**Figure S1**), Zhangjiakou becomes an upstream site in the DFP case (not doubling WFC). Higher $PM_{2.5}$ concentrations are conveyed from the severely contaminated Beijing-Tianjin-Hebei area, and the upstream edge effect (Mo et al., 2017) might slow down the wind speed and, as a result, enhance

$PM_{2.5}$ concentrations.

## 4 Discussions

The fluctuating $PM_{2.5}$ within and around the WFC can be interpreted by atmospheric dynamics and thermodynamics. In addition to the wind farm "edge" effect owing to the horizontal step changes in underlying surface characteristics (Mo et al., 2017), the wind turbine rotors generated turbulence that

could, on the one hand, produce eddies that increase the vertical mixing of momentum and thus, reduce the wind speed at the turbine hub-height level (Baidya Roy, 2004, 2011; Barrie and Kirk-Davidoff, 2010). On the other hand, the increasing vertical mixing could also increase the vertical mixing of scalars such as air temperature and air pollutants. There are statistically significant negative correlations between $\Delta PM_{2.5}$ and $\Delta TKE$ and between $\Delta PM_{2.5}$ and $\Delta V$ (V is wind speed), where $\Delta PM_{2.5}$ indicates the difference

between $PM_{2.5}$ concentration from the WFC-related scenario simulations and the BASE scenario run; the same applies for $\Delta TKE$ and $\Delta V$. **Figure S14** is a correlation diagram between $\Delta PM_{2.5}$ and $\Delta TKE$, and between $\Delta PM_{2.5}$ and $\Delta V$ in January 2016 within and downstream of the WFC, calculated by the differences of these three variables between the DFP and BASE scenario runs. The negative correlations can be seen both inside and outside the WFC, which clearly manifests that growing turbulence and wind

speed from the WFC-related simulations reduced $PM_{2.5}$ concentration compared with the no-WFC (BASE) simulation, and vice versa.



We further estimated vertical cross sections of modelled monthly fractions $PM_{2.5}$, TKE, air temperature, and wind speeds of the three WFC-related scenario runs (SRL, DFP, DOU) to the BASE run in January 2016 along the transect across the WFC and its downstream region, as shown by the solid green line in **Figure S15**, which also illustrates the monthly $PM_{2.5}$ concentration difference between DFP and BASE (=DFP-BASE) in January 2016. Negative fractions can be readily identified in the immediate upstream region and within the WFC, and positive fractions are present in the downstream region, which indicates declining $PM_{2.5}$ in the WFC and the increase of $PM_{2.5}$ levels in the downstream of the WFC (**Figure 2b**). Vertically, we can observe negative $PM_{2.5}$ concentration fractions at a relatively low level of the atmosphere within the WFC from the three WFC-related simulations that extend from the ground surface to approximately 100 m height (**Figure 9a**), which shows that WFC tends to reduce $PM_{2.5}$ compared with its surroundings. Monthly TKE fractions between the three WFC-related model scenario runs and the BASE simulation illustrate a clear positive profile at and above the hub height within the WFC from all three WFC-related scenario simulations (**Figure 9b**), which confirms the increasing turbulence intensity driven by rotated wind turbine rotors. Compared with no-WFC (BASE) case, TKE in the WFC increased up to a factor of 5-6 as shown by the positive TKE fraction of 500%-600% (**Figure 9b1-b3**), and high TKE fractions extend from 150 – 800 m with the maximum at 300-400 m within the WFC. The increasing TKE throughout the ABL within the WFC is in line with Fitch et al.'s modelling results (2012) but their maximum TKE occurred between 100 and 150 m height, and the negative fraction in the hundred kilometre downstream of the WFC at a lower atmospheric level near the surface compared with the positive fraction within the WFC. Our result also illustrates clearly that the negative TKE fractions, indicting decreasing TKE in the presence of the WFC, extend hundred kilometres downstream of the WFC, again agreeing well with Fitch et al.'s simulations (2012). The negative wind speed fractions within the WFC with the maximum negative fractions at the turbine height (100-120 m) and wind speed deficit in the wake region are consistent with Fitch et al. (2012) as well. Both modelling results show negative wind speed differences between windfarm and no-windfarm scenario simulations from windfarms and their downwind regions of several hundred kilometres (**Figure 9c1-c3**). Note also a positive temperature fraction near the surface and the negative temperature fraction centred at the 500-m height, which manifests increasing air temperatures near the surface and decreasing temperature at the relatively high elevation, agreeing to some extent with windfarm modelling result using large eddy simulations (Porté-Agel et al., 2014). The positive temperature fractions, manifesting increasing



temperature, in the downwind region of the wind farms up to a thousand kilometre have been also reported previously (Wang and Prinn, 2010: Baidya Roy and Traiteur, 2010: Vautard et al. 2013; and Sun et al. 2017).

These vertical profiles provide insights into the understanding of the responses of $PM_{2.5}$ to the presence of large-scale wind farms. As previously mentioned, the wind turbine-generated turbulence enhances the vertical mixing of momentum and scalars in the wind farm (Baidya Roy, 2004, 2011), which tends to weaken the momentum near the surface and enhance the momentum above the hub height. The conversion of kinetic energy to mechanical energy further reduces the wind speed at the hub height

(**Figure 9c**). The vertical profiles of air temperature in wind farms have been extensively investigated (Baidya Roy, 2004; Baidya Roy and Traiteur, 2010; Wang and Prinn, 2010; Zhou et al., 2012). These studies have shown marked effects on near surface temperatures from the increased vertical mixing owing to turbulence generated by wind turbine rotors. The increasing temperatures near the surface overlaid by decreasing temperatures aloft (**Figure 9d**) imply a negative lapse rate associated with an

unstable boundary-layer, which leads to stronger vertical mixing, which in turn enhances vertical diffusion of $PM_{2.5}$ and, as a result, reduces concentrations of $PM_{2.5}$ near the surface (**Figure 9a**).

    The negative TKE fractions centred near the southern boundary of the WFC at a relatively lower level compared with its positive fraction within the WFC suggests weakening turbulent activities and mixing that reduces $PM_{2.5}$ vertical mixing and diffusion and thus results in higher $PM_{2.5}$ concentrations

downstream of the WFC (**Figure 9a** and **9d**). The positive $\Delta PM_{2.5}$ vertical profiles indicate enhanced $PM_{2.5}$ concentration approximately 600 km to the south (downstream) of the WFC, as shown in **Figure 9a**, seem to agree to some extent with the perturbations in regional temperature and other weather conditions forced by large-scale wind farm arrays that are hundreds and thousands of kilometres distant from wind farm installations (Barrie and Kirk-Davidoff, 2010; Vautard et al., 2014). A close look at

**Figure 9a** can also identify relatively large values of $\Delta PM_{2.5}$ in the southern boundary and immediately downstream of the WFC. Wang and Prinn (2010) and Keith et al. (2004) have reported that temperature perturbations induced by a large-scale land installation of wind turbines can spread well outside the installation regions. As aforementioned, our results show that both $\Delta V$ and $\Delta T$ profiles within the WFC extend to the downstream of the WFC. The negative $\Delta V$ within the WFC is extended to its southern

boundary and immediate downstream (**Figure 9c**). This can be further identified in **Figure S16**, which shows the differences of monthly wind speed between the DFP and BASE simulations. Negative $\Delta V$ in



the immediate downstream of the WFC can be clearly observed. Among the windfarms in northern China, the area covered by the WFC in Hebei Province alone is more than 5250 km$^2$ and likely forms a long distance wind deficit wake to the downstream. As a result, the weak wind speed favours the accumulation of PM$_{2.5}$ concentrations, characterized by higher PM$_{2.5}$ concentrations. The turbulence activities induced by the steep surface roughness changes might also play an important role. The rough-to-smooth surface transition from the WFC to its downstream leads to the reduction of the friction velocity and turbulent intensity, which could otherwise reduce the vertical mixing of PM$_{2.5}$ concentrations and result in higher PM$_{2.5}$ concentrations near the surface.

## 5 Conclusions

Extensive model sensitivity simulations using the WRF-Chem model were conducted to assess the effect of large-scale wind farms in northern China on PM$_{2.5}$ and air quality forecasts in the NCP, the region most contaminated by PM$_{2.5}$ in China. We quantified the perturbations of hourly and monthly PM$_{2.5}$ concentrations induced by a large-scale WFC in typical winter and summer months across western Inner Mongolia, northern Hebei Province, and the border region between eastern Inner Mongolia and Jilin Province. We show that the WFC tends to increase PM$_{2.5}$ levels in its downstream regions. Our modelling results revealed that the WFC enhanced the PM$_{2.5}$ level in northeast Hebei Province by 49% in the wintertime (January) and 12% in Zhangjiakou in the summertime (July), the places adjacent to the WFC. The model scenario simulations with and without the presence of WFC yielded strong hourly concentration fractions and perturbations, which indicates the marked influences of WFC on the PM$_{2.5}$ environmental fate and forecasting. The WFC more significantly perturbs the PM$_{2.5}$ air concentrations in the wintertime than in the summertime, which is associated with prevailing and local wind fields and with the wind speed deficit wake and "edge" effect. We argue that the wind turbine rotor-generated TKE enhances the vertical mixing of PM$_{2.5}$, which reduces its level near the surface of the WFC. The wind deficit wake and weak TKE extending to the downstream of the WFC might increase PM$_{2.5}$ levels.

The marked changes in PM$_{2.5}$ concentration forced by the WFC in its inside and downstream regions manifest that the influences of large-scale wind farms on air quality forecasts and emissions mitigation should not be overlooked. Severe haze pollution characterized by elevated concentrations of PM$_{2.5}$ in the Beijing-Tianjin-Hebei region has received widespread concern in the Chinese government and the





scientific community. Great efforts have been made to control and reduce PM$_{2.5}$ pollution and to identify

the causes of PM$_{2.5}$ formation. For example, China's State Council issued the "Action Plan on Prevention

and Control of Air Pollution" in 2013, which requests that the most stringent measures are taken to control

haze (SCC, 2013). The Action Plan aimed to reduce PM$_{2.5}$ in the Beijing-Tianjin-Hebei region by 25%

in 2017 compared to 2012 levels. In September 2018, the Ministry of Ecology and Environment of China

issued a new action plan to control air pollution in the Beijing-Tianjin-Hebei region and its surrounding

areas (MEPC, 2018), which requests that further refined management measures be taken to reduce PM$_{2.5}$

levels and to develop more accurate modelling tools to predict the air quality in this region. It is likely

that the WFC and its future expansion under the national clean energy development plan (Dai et al., 2018;

Sahu et al., 2018) would bring additional difficulties in the air quality forecasting and national strategy

to reduce PM$_{2.5}$ in the Beijing-Tianjin-Hebei region. More studies should be conducted to pinpoint the

levels, intensities, and locations of PM$_{2.5}$ perturbations induced by large-scale wind farms.

*Supplement*. The Supplement related to this article is available online.

*Author Contributions*. J.M. and T.H. designed the study. S.L. performed model simulations. J.M. and J.L.,

X.Z., and J.D. collected the measured data to validate the model. J.M., S.L., and T.H. analyzed the data,

interpreted the results, and drafted the manuscript. Other authors participated in the acquisition, analysis,

or interpretation of data.

*Competing Interests*. The authors declare that they have no conflict of interest.

*Acknowledgements*. This work is funded by the National Natural Science Foundation of China through

grant U1806207, 41977357, 41877507, and the National Key R&D Program of China through grant

2017YFC0212002.

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

**Figure Captions**

**Figure 1.** Nested model domains and installed power. d01 is the outer domain entirely covering northern China and d02 is the inner domain covering part of Inner Mongolia, the NCP, and surroundings. The areas with red colour shading highlight major large-scale wind farms (WFC) in northern China. Red pentagram, black squares, and triangles represent Beijing (BIJ), Zhangjiakou (ZJK), and Tianjin (TIJ), respectively. Two blank elliptical circles represent northeast (NHB) and central (CHB) Hebei province.

**Figure 2.** Modelled monthly averaged $PM_{2.5}$ concentration fractions between WFC-related model scenarios and the BASE scenario in January 2016 in the inner model domain. **(a)** Monthly mean concentration fractions between SLR (S2) model run and BASE (S1) simulation; **(b)** same as **Figure 2a** but for the fraction between DFP (S3) and BASE (S1) simulations; **(c)** same as **Figure 2a** but for the fraction between DOU (S4) and BASE (S1) simulations. $PM_{2.5}$ fractions are calculated by $f_i = (C_{Si} - C_{BASE}) \times 100 / C_{BASE}$. The areas where the monthly $PM_{2.5}$ fractions are significant at the 95% confidence level ($t$-test) are highlighted by the black dots. Grey shading areas denote the location of the WFC.

**Figure 3. (a)** WRF-Chem modelled fractions (%) of hourly $PM_{2.5}$ concentrations in January 2016 from SRL simulation to that from the BASE simulation in northeast and central Hebei Province (NHB, CHB), Beijing (BEJ), Tianjin (TIJ), and Zhangjiakou (ZJK). **(b)** Monthly average hourly $PM_{2.5}$ concentrations over January 2016 in the five selected regions and cities. The locations of these five regions and cites are marked in **Figure 1**.

**Figure 4.** Modelled fractions of $PM_{2.5}$ concentration near the surface, the differences of winds (m s$^{-1}$), and fractions of TKE (m$^2$ s$^{-2}$) at the hub height (the fourth level of WRF-Chem) between the DFP and BASE model scenario simulations at 0700 LST January 3 (maximum positive fractions) and 1500 LST January 16 (maximum negative fractions), 2016. **a** and **b**. $PM_{2.5}$ fractions at 0700 LST January 3 (maximum positive fractions) and 1500 LST January 16 (maximum negative fraction), 2016; **c** and **d**, the same as **Figure 4a** and **4b** but for wind speed differences; **e** and **f**, the same as **Figure 4a** and **4b** but for TKE fractions.





**Figure 5.** Modelled monthly averaged $PM_{2.5}$ concentration and concentration differences between WFC-related model scenarios and the BASE scenario in July 2016. **(a)** Monthly mean concentration from BASE (S1) simulation; **(b)** $PM_{2.5}$ concentration differences between SRL (S2) and BASE (S1) simulations; **(c)** same as **Figure 5b** but for the differences between DFP (S3) and BASE (S1) simulations; **(d)** same as **Figure 5b** but for DOU (S4) and BASE (S1) simulations. $PM_{2.5}$ differences are calculated by ($C_{Si} - C_{BASE}$), where $C_{Si}$ denotes modelled concentrations from different model scenarios (i=2, 3, 4). The areas where the monthly $PM_{2.5}$ fractions are significant at the 95% confidence level (*t*-test) are highlighted by the black dots.

**Figure 6.** WRF-Chem simulated hour concentration fractions of $PM_{2.5}$ from model scenario SRL to that from the control run in the five selected regions and cities in July 2016 **(a)**, and monthly averaged concentration fractions in the selected places in July 2016 **(b)**. The geographic location of these five regions are shown in **Figure 1**. Error bars indicate ±1 standard deviation.

**Figure 7.** Modelled $PM_{2.5}$ concentration fractions in daytime and nighttime averaged over January 2016. **(a)** the monthly averaged concentration of $PM_{2.5}$ during the daytime (0700-1800 LST); (b) the monthly averaged concentration of $PM_{2.5}$ during the nighttime (1900-0600 LST); **(c), (d), (e)** the monthly averaged concentration fractions of $PM_{2.5}$ during the daytime between WFC-related (SRL, DFP, DOU) model scenarios and the BASE scenario simulations; **(f), (g), (h)** the same as **Figure 7c-e** but for the nighttime cases. Regions where the $PM_{2.5}$ fractions exhibit 95% confidence level calculated by the contrast experiment *t*-test (Method section) are highlighted with black dots. The grey area indicates the WFC locations.

**Figure 8.** Modelled hourly $PM_{2.5}$ concentration fractions between DOU and DFP scenario runs and the BASE scenario run in Zhangjiakou City in January 2016.

**Figure 9.** Vertical cross-sections of modelled fractions [(DOU-BASE)×100/BASE] of monthly $PM_{2.5}$ concentration (**a**), TKE (**b**), wind speed (**c**), and air temperature (**d**) between the three WFC-related model scenarios of SRL (left panel), DFP (mid panel), and DOU (right panel) and the BASE model scenario run in January 2016 along the transect across the WFC, bounded by the black dashed line, and its



downstream, as shown in the solid green arrow line in **Figure S15**.



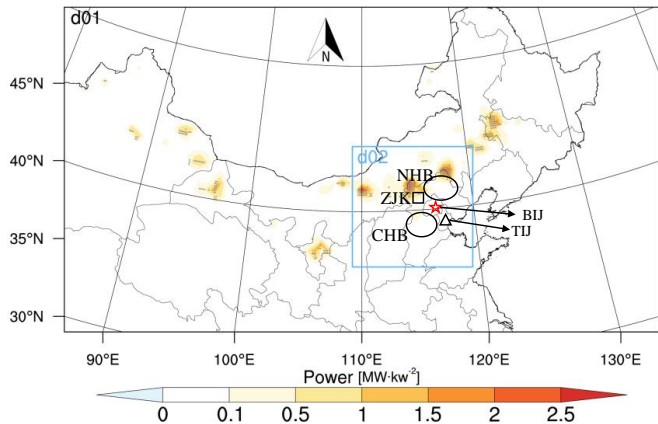

**Figure 1**



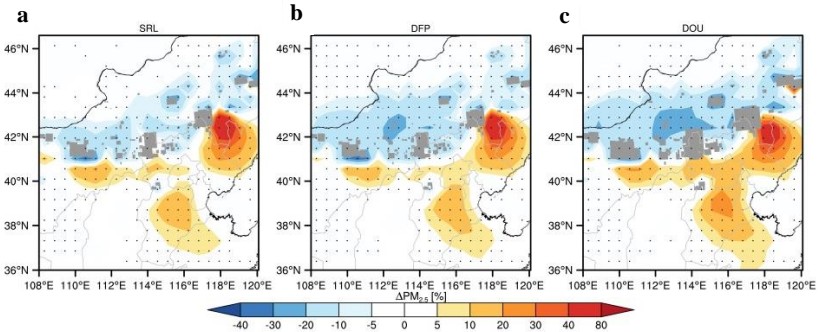

**Figure 2**

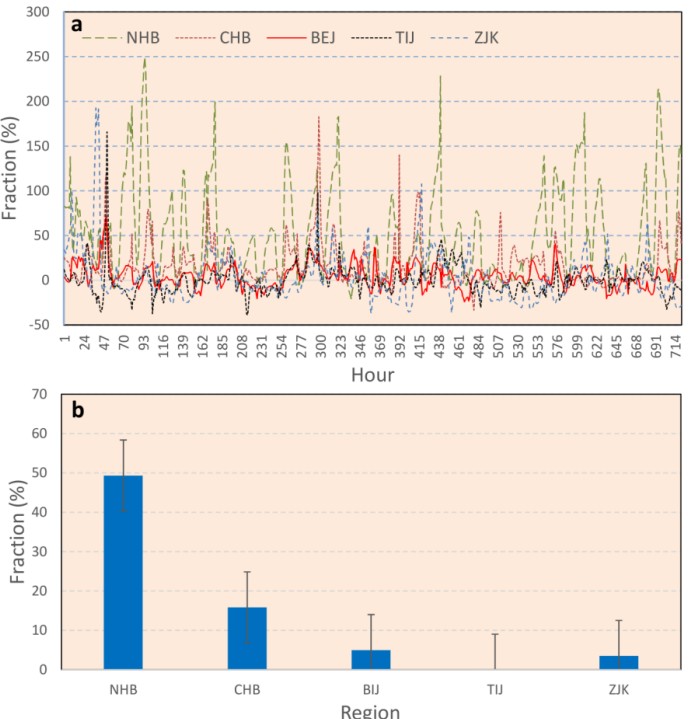

**Figure 3**



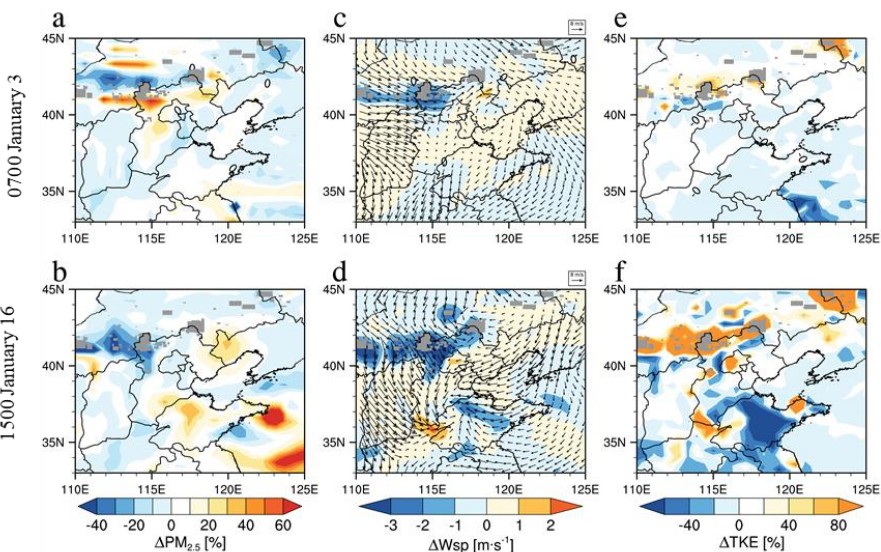

**Figure 4**





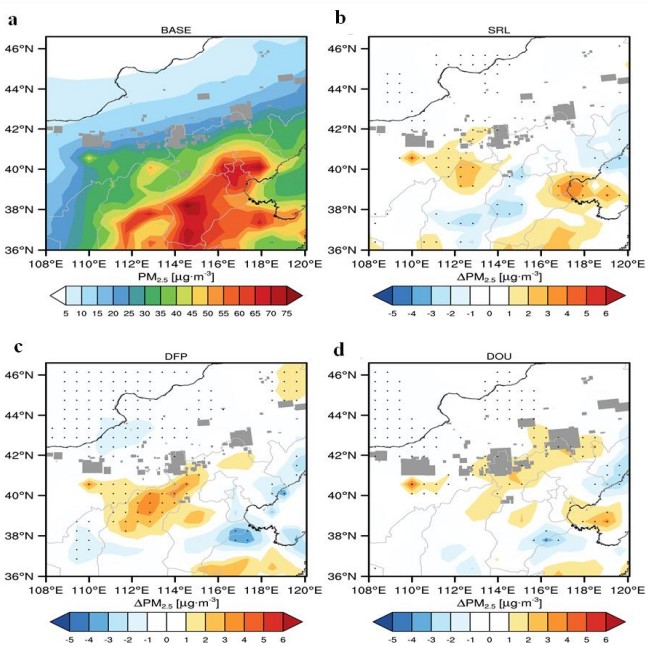

**Figure 5**



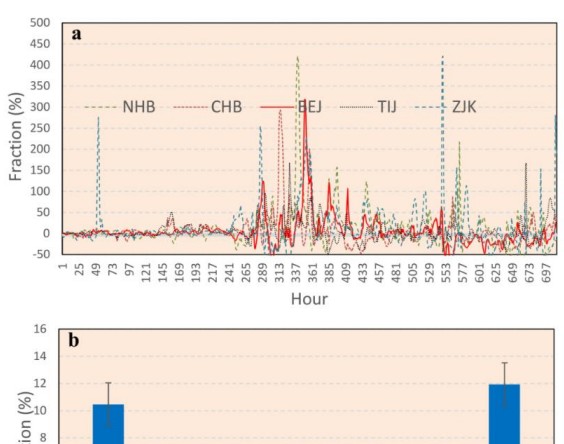

**Figure 6**

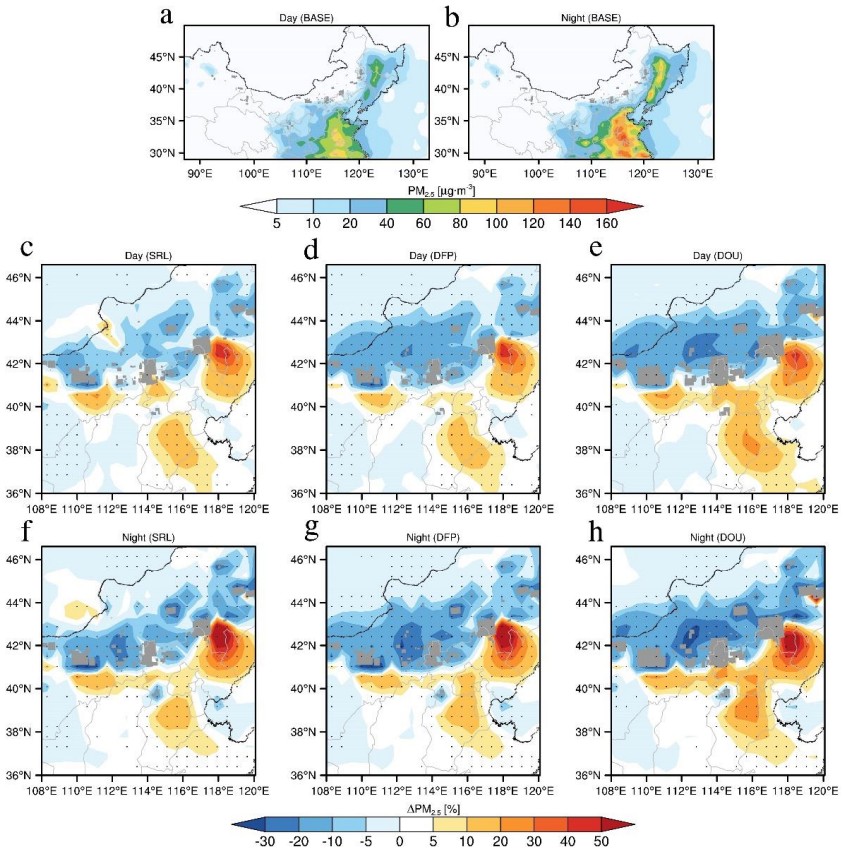

**Figure 7**




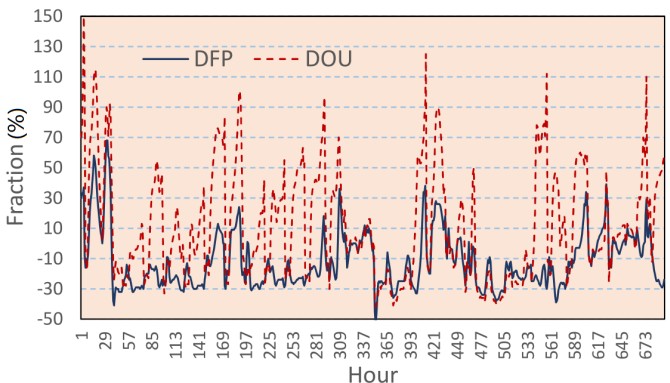

**Figure 8**



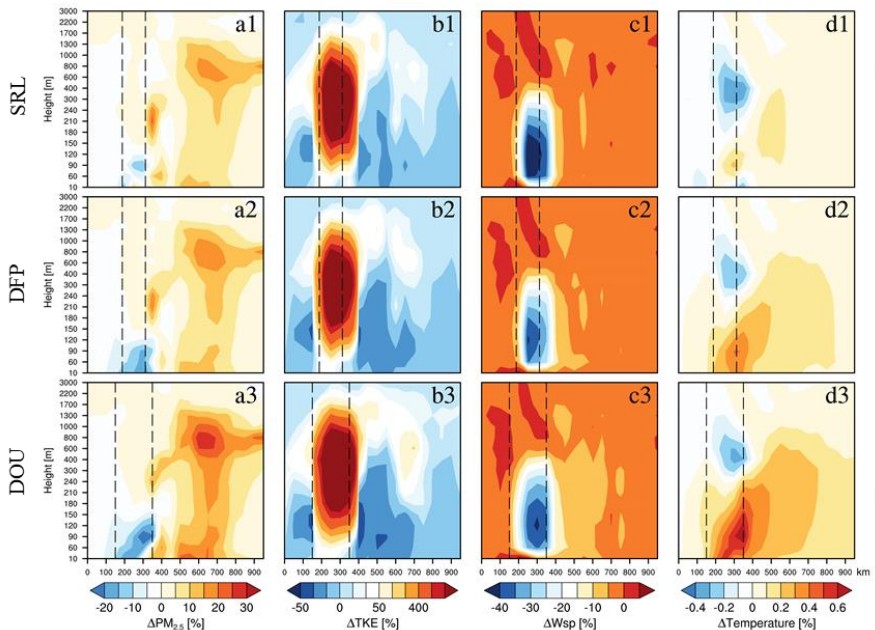

**Figure 9**