# Peer review of "Do large-scale wind farms affect air quality forecast? Modeling evidence in Northern China"

_Atmospheric Chemistry and Physics, 2019_

## Referee Comment (RC1) · Anonymous Referee #3 · 30 Apr 2020

**General Comments**

This study presents a modeling work investigating the influence of large-scale wind farms on surface PM2.5 concentrations. Two wind farm parameterization schemes were adopted in the simulations to simulate the influence of the intensive wind farms over northern China. The changes in surface PM2.5 concentrations induced by the wind farms were then estimated by comparison between simulations with and without wind farm schemes. Theoretically, the wind farms will affect the local and regional meteorological condition, particularly the boundary layer turbulence. However, the analysis and presentation showed in this study did not convince that the wind farms would have significant impacts on regional PM2.5. Further analysis and clarifications are needed before publication.

[Figure]

Major Comments

1. Line 143 of page 7, it is not appropriate to conduct t-test based on hourly data because there is evident diurnal variation of surface PM2.5 concentration. It makes more sense to conduct t-test based on daily mean values. This may be the reason that figures (e.g., Fig. 2) show that the areas are with significance at 95% confidence level but in fact with very small changes. This also raises another concern about the ensemble runs, which is discussed in the comment below.

2. In the Model evaluation section (Text 1 in Supporting Information), only the simulation without wind farm parameterization was compared with measurements. However, in the real world, the intensive large-scale wind farms have been on operation and might have influenced the atmospheres and measurements. Therefore, the simulations with wind farm parameterization schemes should also be validated against measurements in order to demonstrate (1) whether the two schemes could reproduce the impacts of intensive large scale wind farms on the atmospheres and the pollution distribution; (2) whether the simulations with wind farm schemes perform better for reproducing the meteorological fields (e.g., near surface wind, temperature, and PM2.5 concentration) than the simulation without wind farm scheme; (3) the sensitivities and uncertainties of the two wind farm schemes in simulation of the wind field and TKE variation in the domain studied.

3. In terms of evaluation, Fig. S4 seems indicating the model performance is so poor over these 5 stations. Any specific reason? Many studies investigated WRF-Chem simulations of surface PM2.5 concentration over China. Their results are much better than this. Can we really trust the sensitivity analysis based on such model configuration?

4. Many figures used fraction to indicate the impacts. It may be good for some cases, but for the results in this study it may not be appropriate. Fraction sometimes is misleading. For example, in Fig. 2, the area with larger fraction is simply due to its small

concentration instead of due to its large change. The absolute values of concentrations and changes are preferred. The same issue for Fig. 3. These figures show the fraction seems large, however, it is just due to small concentration. Do we really care about the increase from 1 ug/m3 to 2 ug/m3 even though the fraction change is 100% increase? Many places in the text show high fraction value such as 400% in the main text or -40% to 250% in the abstract, which is just misleading. If we look at the absolute values of changes of PM2.5 concentration over the polluted region, mostly less than 10%. This must be revised and clarified.

5. The authors attribute the increase of surface PM2.5 in the downstream of wind farms (Figure 2 and Figure S6) to the decreasing wind speeds (Figure S8) and decreasing turbulent mixing that weaken the vertical mixing and transport of the air pollutants (LINE 277-285). The analysis is not convincing. First, according to Figure S8, the wind speeds reduce in the downstream (south) of the wind farms, and the reduction is about $0.5 \sim 0$ m/s in regions of Beijing-Tianjin and negligible in the central Hebei province. However, the positive fraction of PM2.5 is much higher and more evident in the central Heibei province than that in Beijing-Tianjin, as shown in Figure 2 and Figure S6. Why? Second, wind turbines reduce the hub-height wind speed in downwind areas, at the meantime, generate intensive turbulence in turbine wakes, as the authors cited in LINE 73-74 (Porté-Agel et al., 2014; Li et al., 2018; Fitch et al., 2013; Baidya & Traiteur, 2010; Frandsen et al., 2006). The intensified turbulent strength could enhance the vertical mixing of momentum, heat, moisture and air pollutants. In this sense, the PM2.5 concentrations near the surface in downstream should decrease instead of increasing as shown in this study. Could the authors provide more information and discussion about how meteorological fields in downstream respond to the large-scale wind farms?

6. According to Figure 3a, the monthly average hourly concentration fraction between SLP simulation and BASE simulation is 3% for Zhangjiakou (blue dash line in Figure 3a and LINE 304-308). Figure 8 shows the same information but for the hourly fraction between DFP simulation and BASE simulation. The monthly mean hourly fraction in

Figure 8 is -12.0% for Zhangjiakou (black solid line in Figure 8 and LINE 486). It seems that there may be large difference between the simulations with the two wind farm schemes (3% in SLP vs -12.0% in DFP). Could the authors evaluate the possible difference between the two wind farm schemes?

7. Another major concern is about the case analysis without ensemble runs. For example, Fig. 4, Fig. 6, and Fig. 8. Without ensemble runs, i.e., comparison with one single continuous run with another, it may be fine to compare their monthly mean fields because we can take each day as one ensemble member. However, it doesn't make sense to compare the results at a particular day, because the difference between the two simulations can be simply due to the chaotic signals. If the case analysis is needed, the ensemble runs are necessary.

8. Fig. 5d shows the double of wind farms reduce the overall impacts, why?

9. Figure S10 demonstrates the correlations between $\Delta$PM2.5 and $\Delta$TKE, and between $\Delta$PM2.5 and $\Delta$V in January 2016 within and downstream of the WFC. Could the authors provide more analysis of the correlation between $\Delta$TKE and $\Delta$V within and downstream of the WFC? This may be helpful in depicting the changes induced by the intensive large-scale wind farms on atmospheric dynamics and thermodynamics.

Minor Comments

1. Many figures (e.g., Table S2, Fig. S6, Fig. S8) are discussed extensively but are put in the supporting material, which is not convenient for readers.

2. Fig. 9, SRL is top panel, DOU is bottom panel. Need correction.

3. Line 645-650, these statements are not convincing based on the analysis in this study.

4. In the summer case, "Positive $\Delta$PM2.5 can be discerned in the south (downstream) of the WFC, . . ." (LINE 396). The prevailing wind in northern China in summer consists of mainly southerly and southeasterly. So "downstream" should be "upstream"?

---

## Referee Comment (RC2) · Anonymous Referee #2 · 19 May 2020

This study quantifies the impact of wind farms in northern China on PM2.5 concentrations in the North China Plain using the WRF-Chem model, which could potentially advance our knowledge of the air quality impact of wind power generation. This version of the manuscript has been improved to some extent compared with the last version I reviewed. The authors have addressed some of my major comments. However, I think more evidence is still needed to support their main findings.

1. I appreciate that the author added an evaluation against meteorological observations. However, the present evaluation results are obviously not enough for the purpose of this study. First, the author only evaluated the BASE scenario without wind farms. In fact, the scenario with wind farms included is supposed to be more close to real-world situations. The author needs to compare simulations in all model scenarios

with observations to examine whether the simulated impact of wind farms is in line with observations. Second, the authors just used five selected met observational stations, while NCDC has hundreds of (or at least tens of) stations in the modeling domain which should be included in comparison. Also, the stations the author selected are far away from the wind farms, I suggest that the author specially look into some stations close to the wind farms to see if the simulated perturbation of meteorology by the wind farms is consistent with observed patterns.

2. The authors have also added a comparison of their simulated meteorology perturbation by the wind farms with a couple of previous studies. However, the current comparisons are all qualitative (either increase or decrease). Can the authors do some more quantitative comparisons to examine if the magnitude of meteorological perturbation in their simulations are roughly consistent with previous work? Although different studies are looking at different locations and time periods, I think the perturbation should at least be within the same order of magnitude. If a larger difference is found, I would expect a reasonable explanation why this happens.

3. The authors only conducted simulations in two months, which weakens the robustness of the conclusions given the large variability of the simulation results. The author at least needs to highlight that the magnitude of the wind farms' impact might be quite different for other years or time periods.

4. Why does the model show a large decrease in PM2.5 concentrations at the locations of wind farms in winter but nearly no change in summer? Is it attributed to the local atmospheric circulations you mentioned in Section 3.2? I am a bit surprised if the local circulation fully counteracts the wind farm's influence.

5. In the abstract, the author only described the results in winter, which bias readers' understanding because the roles of wind farms are so different in winter and summer. I suggest that the author include both seasons to give a complete and unbiased picture of the wind farms' impacts.

6. The authors mentioned in Line 72 that "The total number of wind turbines in the outer domain (northern China, Figure 1) was approximately 81,000". Is this calculated from the wind farm area and average wind turbine spacing set in your model? Line 77: Which year is the number 72% for?

Technical corrections: 1. Figure 3a, Figure 6a, Figure 8: Please use date instead of hours for the X-axis. 2. Line 65: WFC has already been defined before.

---

## Referee Comment (RC3) · Anonymous Referee #1 · 21 May 2020

Line 151 to 161: These sentences are redundant. Please remove these lines because they are from Fitch et al., (2012).

---

## Author Comment (AC1) · 23 Jun 2020

First of all, we would like to thank the Referee #3 for his/her comments and suggestions to this manuscript which help us to considerably improve the revised paper. We have made efforts to revise our manuscript following the Referee # 3's comments and address all comments from the Referee #3. The presented below are our point-by-point responses to the Referee # 3' comments.

Referee comment: 1. Line 143 of page 7, it is not appropriate to conduct t-test based on hourly data because there is evident diurnal variation of surface PM2.5 concentration. It makes more sense to conduct t-test based on daily mean values. This may be the reason that figures (e.g., Fig. 2) show that the areas are with significance at 95%

confidence level but in fact with very small changes. This also raises another concern about the ensemble runs, which is discussed in the comment below.

Response: Following the Referee # 3 comment, we have replaced t-test of hourly mean PM2.5 by the t-test of daily mean concentrations in revised Figs 2, 5, and 7. We agree with the Referee #3 that the areas with 95% significance level exhibited small differences between the wind farm related and BASE simulations in most places. But this is expected because these scenario runs were based on the same meteorology and precursor emissions of PM2.5. This point has been added to the revised manuscript. Also, as we mentioned in the paper, "the maximum PM2.5 difference was as high as 14 $\mu$g m-3 in highly polluted Beijing-Tianjin-Hebei region, which was a maximum 10% increase in monthly mean PM2.5 concentration."

2. In the Model evaluation section (Text 1 in Supporting Information), only the simulation without wind farm parameterization was compared with measurements. However, in the real world, the intensive large-scale wind farms have been on operation and might have influenced the atmospheres and measurements. Therefore, the simulations with wind farm parameterization schemes should also be validated against measurements in order to demonstrate (1) whether the two schemes could reproduce the impacts of intensive large scale wind farms on the atmospheres and the pollution distribution; (2) whether the simulations with wind farm schemes perform better for reproducing the meteorological fields (e.g., near surface wind, temperature, and PM2.5 concentration) than the simulation without wind farm scheme; (3) the sensitivities and uncertainties of the two wind farm schemes in simulation of the wind field and TKE variation in the domain studied.

Response: we further validated modeled meteorology and PM2.5 concentrations using the two wind farm parametrization schemes against measurements at selected sampling sites. Results are presented in Tables S12 to S17. As expected, the meteorology and PM2.5 derived from the two schemes do not differ significantly with the BASE simulation. Overall, our results show that the DFP scheme yield slightly better prediction for meteorology and PM2.5 at the sampling sites proximate to the wind farms but slightly worse at the observational stations in megacities which are far away from the wind farms, such as Beijing, as compared with the measured data. These statements have been added to the revised section 2.3.

We further accessed the sensitivities and uncertainties of the two wind farm schemes in simulated wind field, TKE, and PM2.5. Detailed results are presented in new SI Text 2 and Table S18. We also added a new paragraph in section 2.3 Model evaluation and uncertainty analysis to introduce briefly the results from sensitivity and uncertainty analyses.

3. In terms of evaluation, Fig. S4 seems indicating the model performance is so poor over these 5 stations. Any specific reason? Many studies investigated WRF-Chem simulations of surface PM2.5 concentration over China. Their results are much better than this. Can we really trust the sensitivity analysis based on such model configuration?

ResponseïijŽIn general, we feel that modeled PM2.5 agreed reasonably well with the measurements, particularly in the wintertime (January), as shown in Fig. S4, except in two megacities Tianjin and Beijing. As we mentioned in the paper, the inner model domain resolution was set 10 km, the modeled gridded PM2.5 cannot match with the sampling sites. The measured PM2.5 in Beijing and Tianjin was averaged over 10-12 sampling sites across the city. In reality, complex urban geometry and precursor sources cause large spatial gradients of PM2.5 distribution, leading to the mismatch between modeled and measured PM2.5 concentrations.

4. Many figures used fraction to indicate the impacts. It may be good for some cases, but for the results in this study it may not be appropriate. Fraction sometimes is misleading. For example, in Fig. 2, the area with larger fraction is simply due to its small concentration instead of due to its large change. The absolute values of concentrations and changes are preferred. The same issue for Fig. 3. These figures show the fraction

seems large, however, it is just due to small concentration. Do we really care about the increase from 1 ug/m3 to 2 ug/m3 even though the fraction change is 100% increase? Many places in the text show high fraction value such as 400% in the main text or -40% to 250% in the abstract, which is just misleading. If we look at the absolute values of changes of PM2.5 concentration over the polluted region, mostly less than 10%. This must be revised and clarified.

Response: we partly agree with the Referee #3's comment. Fraction is widely used in multiple scenario sensitivity analysis. In our study, the fraction was calculated by (CSi – CBASE)×100/CBASE, for a small PM2.5ãĂĂconcentration from the BASE simulation and large concentration from a scenario run, we would expect a large fraction. But this does suggest a strong perturbation induced by the wind farm, even PM2.5 level is low. Attached figure shows the hourly concentration differences ($\Delta$C=Csi – CBASE) and fractions (=$\Delta$C×100/CBASE) between SRL and BASE simulation scenario in Northeast Hebei. As seen, the fluctuations and magnitude of concentration fractions matches well with that of concentration differences, indicating no marked distinctions between concentration fraction and difference from two model scenario simulations. Again this is expected because PM2.5 concentrations from the SRL simulation were predicted subject to the same meteorology and precursor emissions of PM2.5 as the BASE simulation. Hence, we still used the concentration fractions in Fig. 3, 6, and 8. As shown, this does not change the spatial pattern as illustrated by concentration fractions (see Response Fig. 1). However, following the Referee #3's comment, we replaced fractions by concentration differences in Figs. 2, 5, and 7, and in text as possible as we can. For example, we replaced concentration fractions of 192.7% and -37.3% by concentration differences of 27.3 and -6.4 $\mu$g/m3 in Zhangjiakou case study (line 284-287).

5. The authors attribute the increase of surface PM2.5 in the downstream of wind farms (Figure 2 and Figure S6) to the decreasing wind speeds (Figure S8) and decreasing turbulent mixing that weaken the vertical mixing and transport of the air pollutants (LINE 277-285). The analysis is not convincing. First, according to Figure S8, the wind speeds reduce in the downstream (south) of the wind farms, and the reduction is about 0.5 âĹij 0 m/s in regions of Beijing-Tianjin and negligible in the central Hebei province. However, the positive fraction of PM2.5 is much higher and more evident in the central Heibei province than that in Beijing-Tianjin, as shown in Figure 2 and Figure S6. Why? Second, wind turbines reduce the hub-height wind speed in downwind areas, at the meantime, generate intensive turbulence in turbine wakes, as the authors cited in LINE 73-74 (Porté-Agel et al., 2014; Li et al., 2018; Fitch et al., 2013; Baidya & Traiteur,2010; Frandsen et al., 2006). The intensified turbulent strength could enhance the vertical mixing of momentum, heat, moisture and air pollutants. In this sense, the PM2.5 concentrations near the surface in downstream should decrease instead of increasing as shown in this study. Could the authors provide more information and discussion about how meteorological fields in downstream respond to the large-scale wind farms?

Response: perhaps we didn't specify clearly. Figures S6 and S8 (Fig. S9 in the revised SI) are not comparisons of wind speeds and PM2.5 concentrations between the wind farm related simulations and the BASE simulation results. Rather, these two figures present the comparison between modeled and measured air temperatures (Fig. S6) and wind speeds (Fig. S8, or Fig. S9 in the revised SI) at the several sampling sites. In fact, the modeled wind speed reductions in the wind farm simulations were greater than 0.5 m/s at most model grids in the downstream central Hebei province and Beijing-Tianjin. In Fig. 9, the wind speed reductions, as shown by negative wind speed fractions between wind farm related scenario and BASE scenario simulated wind speed, extended from the wind farm to several hundred kilometers downstream up to 600-800 m height. In particular, the doubling wind farm (DOU) simulation leads to 5-10% wind speed reduction in about 600 km downstream from 10 m to 600 m height. This is in line with Vautard et al.'s result (2014). Second, the intensified turbulence occurs in turbine wake. However, in the downstream of the wind farm, as illustrated in Fig. 9b1-b3, TKE was decreased up to 600 km downstream of the WFC below 200-400 m height

as shown by negative TKE fractions. Above the 200-400 m height, we can observe increasing TKE. Following the Referee #3's comment, we added a new Fig. S19 which illustrates wind speed and TKE vertical profiles at a 50 km downstream model grid (20, 53), which could be considered as the turbine wake distance for a large-scale wind farm, and 300 km downstream grid (15, 53), respectively. As seen, the TKE is stronger in the wind farm case than the case without the wind farm at the 50 km downstream grid but weaker at the 300 km downstream. Whereas, the wind speeds in the presence of the wind farm are weaker than the case without the presence of the wind farm. As we mentioned in the paper, the weaker winds leads to accumulation of PM2.5, showing higher concentration in the downstream regions. The corresponding discussions have been added to the revise paper (line 518-525).

6. According to Figure 3a, the monthly average hourly concentration fraction between SLP simulation and BASE simulation is 3% for Zhangjiakou (blue dash line in Figure 3a and LINE 304-308). Figure 8 shows the same information but for the hourly fraction between DFP simulation and BASE simulation. The monthly mean hourly fraction in Figure 8 is -12.0% for Zhangjiakou (black solid line in Figure 8 and LINE 486). It seems that there may be large difference between the simulations with the two wind farm schemes (3% in SLP vs -12.0% in DFP). Could the authors evaluate the possible difference between the two wind farm schemes?

Response: The referee raised an interesting question! The both wind farm schemes have their respective advantages and disadvantages. Previous studies have examined the differences between the DFP scheme and the enhanced roughness length approach, such as the SLP scheme. Fitch et al. (2013) found that these two approaches always exhibited the opposite wake structure. The SLP scheme tends to produce excessive surface fluxes and production of TKE, whereas the DFP scheme tends to overestimate the wake effect (Lee and Lundquist, 2017). Located immediate downstream of the WFC, Zhangjiakou can be considered as a wake region of large scale WFC. Hence, the overestimation of the wake effect featured by wind deficit led

to the relatively large negative wind speed fractions in DFP run compared to the BASE simulation. This point has been added to the revised paper (line 410-414).

7. Another major concern is about the case analysis without ensemble runs. For example, Fig. 4, Fig. 6, and Fig. 8. Without ensemble runs, i.e., comparison with one single continuous run with another, it may be fine to compare their monthly mean fields because we can take each day as one ensemble member. However, it doesn't make sense to compare the results at a particular day, because the difference between the two simulations can be simply due to the chaotic signals. If the case analysis is needed, the ensemble runs are necessary.

Response: we agree with the Referee #3 that ensemble runs might provide better prediction of the winds, air temperature, and TKE, particularly when the initial conditions and different physical parameterizations exhibit errors and uncertainties. However, for PM2.5, it is not straightforward to conduct ensemble simulations given its non-linear responses to precursor emissions and complex chemistry. To our knowledge, presently, the successful operational ensemble forecasts for air quality are essentially based on the combination of multiple air quality forecast models (e.g., Petersen et al., 2019), or combined air quality forecast model with satellite AOD, deep learning prediction, and other statistical methods (e.g., Shtein et al., 2020). We have tried to use a weighted method to perform a simple ensemble prediction of PM2.5 by combining SRL and DFP simulations. The results show that the weighted ensemble simulation didn't improve PM2.5 simulations compared with single SRL and DFP modeling results, likely due to small ensemble numbers (2). Hence, at this time we are not able to carry out ensemble runs for PM2.5. However, following the Referee #3's comment, in the revised paper we have added a new paragraph (the last paragraph of section 2.2,line 168-177) which recognizes this problem and potential errors of scenario modeling results for those particular days with maximum positive and negative fractions between WRC related scenario runs and BASE simulation, as shown in Figs. 3, 6, and 8. We also removed the maximum positive and negative hourly PM2.5 concentrations throughout

the revised paper, e.g., "from -40% to 250%" in Abstract.

8. Fig. 5d shows the double of wind farms reduce the overall impacts, why?

Response: In reality, doubling WFC did not reduce the overall impacts. As shown in Fig. 5d, doubling WFC led to positive concentration differences within and surrounding the WFC, as compared with SRL and DFP simulations (Fig. 5b and c). As we discussed in the paper, "overall, the monthly PM2.5 concentration fractions from the SRL, DFP, and DOU model runs to that of the control run are small. Compared with the January case (Figure 2), no statistically significant positive concentration fractions were identified in the downstream of the WFC in July. This is likely attributed to stronger local boundary layer circulation occurring in the summertime (Garratt, 1994)."

9. Figure S10 demonstrates the correlations between $\Delta$PM2.5 and $\Delta$TKE, and between $\Delta$PM2.5 and $\Delta$V in January 2016 within and downstream of the WFC. Could the authors provide more analysis of the correlation between $\Delta$TKE and $\Delta$V within and downstream of the WFC? This may be helpful in depicting the changes induced by the intensive large-scale wind farms on atmospheric dynamics and thermodynamics.

Response: Following the Referee #3's comment, we added a new Fig. S17 (see Response Fig. 2) in the revised SI showing correlations between modeled $\Delta$TKE and $\Delta$U within the WFC and 300 km downstream and referred in main text (the end of the first paragraph of Discussion section, line 441-443).

Minor Comments 1. Many figures (e.g., Table S2, Fig. S6, Fig. S8) are discussed extensively but are put in the supporting material, which is not convenient for readers.

Response: Table S2 presents the 4 model scenarios. In the revised paper, this Table is not referred extensively anymore and cited once only. Figures S6 and S8 illustrate model evaluations which were not discussed extensively. In fact, most figures in Supplementary, e.g., Figures S4-S9 in previous SI, and S4-S11 in the revised Supplementary show model evaluation results. These results were not discussed extensively as

well, but just referred simply in section 2.3. 2. Fig. 9, SRL is top panel, DOU is bottom panel. Need correction. Response: Corrected, thanks! 3. Line 645-650, these statements are not convincing based on the analysis in this study. Response: We are not sure about the statements questioned by the Referee #3. Line 645-650 are references. 4. In the summer case, "Positive $\Delta$PM2.5 can be discerned in the south (downstream) of the WFC, ..." (LINE 396). The prevailing wind in northern China in summer consists of mainly southerly and southeasterly. So "downstream" should be "upstream"? Response: Yes the Referee is right! "downstream" has been replaced by "upstream".

[Figure]

**Fig. 1.** PM2.5 concentration fractions and differences from the SRL simulation in January 2016

[Figure]

**Fig. 2.** Correlation diagrams of modeled differences of TKE ($\Delta$TKE, m2 s-2) and the wind speed ($\Delta$WSP, m s-1) near the surface between DFP and BASE simulations

---

## Author Comment (AC2) · 23 Jun 2020

Responses to Referee #2

This study quantifies the impact of wind farms in northern China on PM2.5 concentrations in the North China Plain using the WRF-Chem model, which could potentially advance our knowledge of the air quality impact of wind power generation. This version of the manuscript has been improved to some extent compared with the last version I reviewed. The authors have addressed some of my major comments. However, I think more evidence is still needed to support their main findings.

Response: We thank the Referee #2 for his/her comments and suggestions to this manuscript which help us to considerably improve the revised paper. We have made

major revisions to our manuscript following the Referee # 2's comments and address all comments from the Referee #2. The presented below are our point-by-point responses to the Referee # 2' comments.

1. I appreciate that the author added an evaluation against meteorological observations. However, the present evaluation results are obviously not enough for the purpose of this study. First, the author only evaluated the BASE scenario without wind farms. In fact, the scenario with wind farms included is supposed to be more close to real-world situations. The author needs to compare simulations in all model scenarios with observations to examine whether the simulated impact of wind farms is in line with observations. Second, the authors just used five selected met observational stations, while NCDC has hundreds of (or at least tens of) stations in the modeling domain which should be included in comparison. Also, the stations the author selected are far away from the wind farms, I suggest that the author specially look into some stations close to the wind farms to see if the simulated perturbation of meteorology by the wind farms is consistent with observed patterns.

Response: Following the Referee #2's suggestions, we collected measured PM2.5 concentrations, and the winds and air temperature data from additional 3 monitoring sites to evaluate modeling results. Among the 8 sites, four sites are proximate to the WFC, these are Chengde, Siping, Ligong, and Baotou (Table S3). New figures and Tables are provided further evaluations of modeled PM2.5 concentrations and the winds and air temperatures. Now the revised Supplementary includes 8 figures and 14 tables showing the model evaluation results. We also validated modeled meteorology and PM2.5 concentrations from the two wind farm parametrization schemes against measurements at the 5 sampling sites. Results are presented in Tables S12 to S17. As expected, the meteorology and PM2.5 derived from the two schemes do not differ significantly with the BASE simulation. Overall, our results show that the DFP scheme yield slightly better prediction for meteorology and PM2.5 at the sampling sites proximate to the wind farms but slightly worse at the observational stations in megacities

which are far away from the WFC, such as Beijing, as compared with the measured data. These statements have been added to the revised section 2.3.

2. The authors have also added a comparison of their simulated meteorology perturbation by the wind farms with a couple of previous studies. However, the current comparisons are all qualitative (either increase or decrease). Can the authors do some more quantitative comparisons to examine if the magnitude of meteorological perturbation in their simulations are roughly consistent with previous work? Although different studies are looking at different locations and time periods, I think the perturbation should at least be within the same order of magnitude. If a larger difference is found, I would expect a reasonable explanation why this happens.

Response: Following the Referee #2's suggestions, in the revised manuscript we have extended significantly Discussion section by inputting detailed comparisons of meteorological conditions (winds, TKE, and temperatures) between the results from our studies and previous results (2nd paragraph of Discussion section, line 444-492). We added two new figures (Fig. S17 and S19) in the revised Supplementary. Although the magnitude of windfarm induced changes in these met variables differ somewhat from previous results due to different scale and installation of wind farms, and meteorological conditions (spatial scales), we show that these modeled meteorological variables and their downwind distribution agree reasonably well with the previous results. We have made major revisions to this section.

3. The authors only conducted simulations in two months, which weakens the robustness of the conclusions given the large variability of the simulation results. The author at least needs to highlight that the magnitude of the wind farms' impact might be quite different for other years or time periods.

Response: Following the Referee #2's comment, we have added statements "Given large seasonal, inter-annual, and intra-annual variabilities of meteorology and climate, the results and conclusions from the present study might not be applicable in other

years. Further extensive investigations of the influences of wind farms with different scales and installations on air quality are needed." (section 2, line 164-167)

4. Why does the model show a large decrease in PM2.5 concentrations at the locations of wind farms in winter but nearly no change in summer? Is it attributed to the local atmospheric circulations you mentioned in Section 3.2? I am a bit surprised if the local circulation fully counteracts the wind farm's influence.

Response: Observational data show that the mean wind speed in north China in winter under the East Asian winter monsoon are stronger than summer months. Compared with large-scale winter circulations, small-scale summer atmospheric circulations are forced, to a large extent, by local surface heating and cooling. In the revised manuscript, we have added a new reference (Feng et al., 2020) to support this argument (line 346).

4. In the abstract, the author only described the results in winter, which bias readers' understanding because the roles of wind farms are so different in winter and summer. I suggest that the author include both seasons to give a complete and unbiased picture of the wind farms' impacts.

Response: We have added new statements in the revised Abstract.

6. The authors mentioned in Line 72 that "The total number of wind turbines in the outer domain (northern China, Figure 1) was approximately 81,000". Is this calculated from the wind farm area and average wind turbine spacing set in your model? Line 77: Which year is the number 72% for?

Response: The number of wind turbines at 81,000 is calculated by total power capacity 121,500 MW divided by nominal power 1.5 MW of each turbine. 72% is for 2015. This has been specified in the revised paper (line 76-80).

Technical corrections: 1. Figure 3a, Figure 6a, Figure 8: Please use date instead of hours for the X-axis. 2. Line 65: WFC has already been defined before.

Response: Done!
* * *

---

## Author Comment (AC3) · 23 Jun 2020

Referee #1 comments: Line 151 to 161: These sentences are redundant. Please remove these lines because they are from Fitch et al. (2012).

ResponseïijŽWe agree! Following the Referee #1 comment we have deleted these sentences in the revised paper.

––––––––––––––––––––––––